# The Cultivation of *Spirulina maxima* in a Medium Supplemented with Leachate for the Production of Biocompounds: Phycocyanin, Carbohydrates, and Biochar

Wallyson Ribeiro dos Santos [1],*, Matheus Lopes da Silva [1], Geronimo Virginio Tagliaferro [1], Ana Lucia Gabas Ferreira [2],* and Daniela Helena Pelegrine Guimarães [1]

[1]   Chemical Engineering Department, Engineering School of Lorena, University of São Paulo, Lorena 12602-810, Brazil; matheuslopes@usp.br (M.L.d.S.); tagliaferro@usp.br (G.V.T.); dhguima@usp.br (D.H.P.G.)

[2]   Department of Basic and Environmental Sciences, Engineering School of Lorena, University of São Paulo, Lorena 12602-810, Brazil

*   Correspondence: wallyson@alumni.usp.br (W.R.d.S.); gabas@usp.br (A.L.G.F.); Tel.: +55-12-99232-5102 (W.R.d.S.)

**Abstract:** Cyanobacteria are microorganisms that grow rapidly in an aquatic medium, showing the capacity of accumulations of biocompounds subsequently converted into value-added biocompounds. The cyanobacterium *Spirulina maxima* can produce pigments besides accumulating significant amounts of carbohydrates and proteins. An alternative to reducing biomass production costs at an industrial scale is the use of landfill leachate in the growing medium, as well as the mitigation of this pollutant. The objective of this work was to cultivate *Spirulina maxima* in a medium supplemented with leachate, using the design of experiments to evaluate the effects of leachate concentration (% *v/v*), light source, and light intensity in an airlift photobioreactor, analyzing them as a response to the productivity of biomass, phycocyanin, carbohydrates, and biochar. The highest values of productivity (mg $L^{-1}d^{-1}$) were 97.44 ± 3.20, 12.82 ± 0.38, 6.19 ± 1.54, and 34.79 ± 3.62 for biomass, carbohydrates, phycocyanin, and biochar, respectively, adjusted for experiment 2 with the factors of leachate concentration (5.0% *v/v*), light source (tubular LED), and luminosity (54 μmol $m^{-2}$ $s^{-1}$), respectively. The use of leachate as a substitute for macronutrients in Zarrouk's medium for the cultivation of *Spirulina maxima* is a viable alternative in the production of biocompounds as long as it is used at an appropriate level.

**Keywords:** *Spirulina maxima*; carbohydrate; phycocyanin; biochar; leachate; LED

## 1. Introduction

Bioenergy plays a crucial role in the transition to a more sustainable, low-carbon economy. In this context, microalgae and cyanobacteria have emerged as promising sources of biomass for the production of bioproducts and biofuels [1]. Due to their ability to efficiently photosynthesize, these microorganisms can convert carbon dioxide and nutrients into biomass with a high growth rate, making them attractive resources for renewable energy production [2].

*Spirulina* have become popular in the health food industry, being included in diets as protein and vitamin supplements [3]. Despite its low lipid content, this cyanobacterium presents a high percentage of proteins (60% to 70%), which is noteworthy for biomass pyrolysis. This leads to charcoal of comparable quality to lignocellulosic biomass, yet it necessitates a milder temperature [4,5]. Furthermore, *Spirulina* can accumulate carbohydrates and sugars, which can be converted into bioethanol and produce value-added compounds such as pigments and drugs [6,7].

Alternative cyanobacteria cultivation media have gained increasing interest due to the need to develop more efficient and sustainable methods for the production of these

biotechnologically important microorganisms [8]. Traditionally, cyanobacteria are grown in liquid or solid media that require specific nutrients, such as nitrates, phosphates, and trace elements, as well as controlled pH, temperature, and lighting conditions. However, these conventional means present challenges in terms of cost, raw-material availability, and environmental impact [9].

A promising alternative is cultivation media based on agroindustrial residues, such as vinasse, sugarcane bagasse, rice husks, and effluents from industrial processes [10,11]. These organic wastes provide essential nutrients for the growth of cyanobacteria and can be an economical and sustainable source of raw materials for large-scale production [12]. Furthermore, the use of agroindustrial waste contributes to reducing the environmental impact associated with its inadequate disposal, transforming it into valuable resources for the production of microbial biomass [13].

Another innovative approach involves growing cyanobacteria in photobioreactor systems integrated into wastewater treatment processes [14]. These combined systems allow for the removal of nutrients, such as nitrogen and phosphorus, from wastewater while promoting the growth of cyanobacteria. In this way, it is possible to obtain high-quality cyanobacteria biomass while efficiently treating effluents, contributing to the recovery of resources and reducing water pollution [15]. Zarrouk's growing medium is widely used for growing *Spirulina*. However, the compounds used in this cultivation medium can increase the cost of production on an industrial scale [16,17].

Thus, in this work, the proposal was to replace the macronutrients ($NaHCO_3$, $K_2HPO_4$, $NaNO_3$) from Zarrouk's medium (Table S1, Supplementary Material) with leachate (landfill leachate), an environmentally polluting residue, coming from Zarrouk landfill in the municipality of Cachoeira Paulista (initially containing 14.13 mg mL$^{-1}$ of phosphorus; 56.86 mg mL$^{-1}$ of magnesium; 4005.5 mg mL$^{-1}$ of sodium; 2084.5 mg mL$^{-1}$ of potassium; and 36.76 mg mL$^{-1}$ of calcium), which provided the macronutrients necessary for the cultivation of *Spirulina maxima*, resolving the volume of leachate destined for treatment, before being discarded into the environment.

## 2. Materials and Methods

### 2.1. Inoculum Production

The experiments were conducted with the cyanobacterium *Spirulina maxima* (Setchell & N.L.Gardner) Geitler 1932, using a strain provided by the Department of Biological Oceanography of the Oceanographic Institute of the University of São Paulo, located in the city of Ubatuba in the State of São Paulo.

The cyanobacterial strain was kept in a ceparium at the Microalgae Engineering Laboratory (MEL) with a photoperiod of 12 h light/12 h dark. The luminous intensity provided was 10.8 μmol m$^{-2}$ s$^{-1}$ maintained by a 7 W LED (light-emitting diode) lamp. For the maintenance of a cell bank, a subculture was made in 125 mL Erlenmeyer flasks in 15 to 20 days at a proportion of 10 mL of the predecessor culture to 90 mL of new culture medium, and the flasks were shaken manually once a day.

### 2.2. Cultivation of S. maxima in Zarrouk's Medium Modified with Leachate

The experiments were conducted at room temperature with continuous lighting, using leachate from the landfill of the municipality of Cachoeira Paulista, State of São Paulo, which was stored at 5 °C to compose the culture medium. The composition of the leachate is shown in Table 1.

The cultivations were carried out batchwise in airlift photobioreactors starting with an absorbance of approximately 0.400 to maintain the standardization of the initial cell concentration in the experiments. All cultivations were performed in triplicate. The follow-up of the cultures was analyzed using samples taken during the 9 days of cultivation through absorbance readings on a spectrophotometer at a 680 nm wavelength.

**Table 1.** Landfill leachate composition in the municipality of Cachoeira Paulista, SP, Brazil.

| Parameter | Concentration (mg L$^{-1}$) |
|---|---|
| Al | 4.53 |
| P | 141.13 |
| Si | 3.27 |
| Se | 0.17 |
| Cr | 2.3 |
| Mg | 56.86 |
| Na | 4005.5 |
| K | 2084.5 |
| Zn | 1.2 |
| Ti | 0.8 |
| Fe | 6.16 |
| Ca | 36.76 |
| Sn | 1.14 |
| COD (mg L$^{-1}$ O$_2$) | 3565.0 |
| pH | 8.78 |

COD (Chemical Oxygen Demand).

The experiments were carried out by varying the leachate concentration in the medium (% *v/v*), light source, and light intensity (µmol m$^{-2}$ s$^{-1}$) to determine the best cultivation condition of *Spirulina maxima*. The experimental cultures followed Taguchi's [18] L4 experimental design with four independent factors and two levels, shown in Tables 2 and 3.

**Table 2.** Factors and levels of the cultivation assays.

| Codified Factor | Factor | Levels | |
|---|---|---|---|
| | | 1 | 2 |
| A | Leachate concentration (% *v/v*) | 5 | 10 |
| B | Luminous source (lamp) | Fluorescent | Tubular LED |
| G | Luminosity (µmol m$^{-2}$ s$^{-1}$) | 13.5 | 54 |

**Table 3.** Experimental matrix of Taguchi's L4 planning with seven factors and two levels.

| | Factors | | |
|---|---|---|---|
| Experiment | A | B | C |
| 1 | 1 | 1 | 1 |
| 2 | 1 | 2 | 2 |
| 3 | 2 | 1 | 2 |
| 4 | 2 | 2 | 1 |

*2.3. Analytical Methods*

Phycocyanin extraction was initiated by adding 100 mg of dry biomass to a Falcon tube. Subsequently, a solution comprising 12.5 mL of sodium dihydrogen phosphate (NaH$_2$PO$_4$) and sodium chloride (NaCl) buffer at concentrations of 0.01 mol/L and 0.15 mol/L, respectively, was added. The mixture was then subjected to ultrasonication for 30 min at a frequency of 50 kHz. Following this, three freeze/thaw cycles were conducted: the biomass was frozen for 24 h and thawed at 25 °C. After the freeze/thaw cycles, the samples were centrifuged at 3500 rpm for 40 min, leading to the separation of the supernatant containing the pigment. Subsequently, the supernatant was analyzed using a spectrophotometer at wavelengths of 615 nm and 652 nm. The concentration of phycocyanin [19] was defined by the following equation:

$$CF = \frac{OD_{615} - 0.474 \times (OD_{652})}{5.34} \qquad (1)$$

where *CF* is phycocyanin concentration (mg mL$^{-1}$), $OD_{615}$ is the optical density of the sample at 615 nm, and $OD_{652}$ is the optical density of the sample at 652 nm.

The determination of reducing sugars, initiated with the hydrolysis of *Spirulina maxima* biomass, followed the protocol of [20], in which approximately 300 mg of dry biomass was subjected to acid hydrolysis. Initially, the material was treated with 72% sulfuric acid in test tubes and kept in a water bath at 30 °C for 60 min. After hydrolysis, the contents were transferred to a 125 mL Erlenmeyer flask containing 84 mL of distilled water, with the concentration of sulfuric acid then being diluted to 4%. Subsequently, the sample was autoclaved at 121 °C for 60 min and filtered by using a Gooch crucible, and the hydrolysate was stored in a refrigerator.

The DNS method for determining reducing sugars is based on the reaction between reducing sugars and 3,5-dinitrosalicylic acid. First, a standard reducing sugar curve was established using glucose in concentrations ranging from 0.1 g L$^{-1}$ to 1 g L$^{-1}$. The biomass hydrolysate was analyzed by pipetting 1 mL of it into a test tube along with 3 mL of DNS solution. The mixture was heated in a thermostatic bath at 100 °C for 5 min, followed by analysis in a spectrophotometer at 540 nm. The concentration of the reduction groups was calculated by using the calibration curve equation (Figure S1, Supplementary Material) based on the measured absorbance.

The analysis of biochar began with drying the biomass at 40 °C for 48 h to minimize the moisture content. Then, pyrolysis was conducted. Initially, approximately 50 mg of dry biomass was placed in pre-weighed and dried ceramic crucibles. Pyrolysis took place in a muffle furnace with restricted oxygen, ramping up at a rate of 10 °C every 10 min until reaching 310 °C. Subsequently, the biomass underwent calcination for 60 min and was left to cool to room temperature inside the muffle furnace for approximately 24 h. The aim was to obtain Spirulina maxima biochar gradually, ensuring a high solid yield [21].

The biochar yield was determined by the mass ratio between the biochar obtained after pyrolysis and the dry biomass:

$$\% \, biochar = \frac{\left(C_f - C_s\right)}{Bs} \times 100 \tag{2}$$

where $C_f$ is the mass of the crucible containing the biochar after the pyrolysis process, $C_s$ is the mass of the dry crucible, and *Bs* is the mass of dry biomass to be pyrolyzed.

### 2.4. Statistical Analysis

The results obtained were analyzed by using the STATISTICA 13.5 program to verify the effects of the factors (CC—leachate concentration, FL—light source, and Lum—luminosity) on the response variables (PB—biomass productivity, PF—phycocyanin productivity, PA—sugar productivity, and PB—biochar productivity). After data analysis, the software performed an analysis of variance (ANOVA) and effect plotting for each of the factors. The analysis of variance enabled us to determine which factors were significant for the process (*p*-value below 0.05 or 0.10), and the effect graph helped us to observe the best fit for the production of each of the response variables.

## 3. Results and Discussion

### 3.1. Development of Spirulina maxima in Medium Supplemented by a Leachate

Figure 1 illustrates the differences in absorbance (ABS) observed between the cultivation curves. The differences shown in Figure 1 allowed us to observe that the variations in the experimental conditions of the parameters had considerable effects on the growth profile of *Spirulina maxima*. However, a linear growth profile occurred in all experiments, with no identification of the lag phase or exponential phase of growth in the batch. The absence of an exponential phase indicates that the assimilation of leachate can be difficult in metabolization by cells.

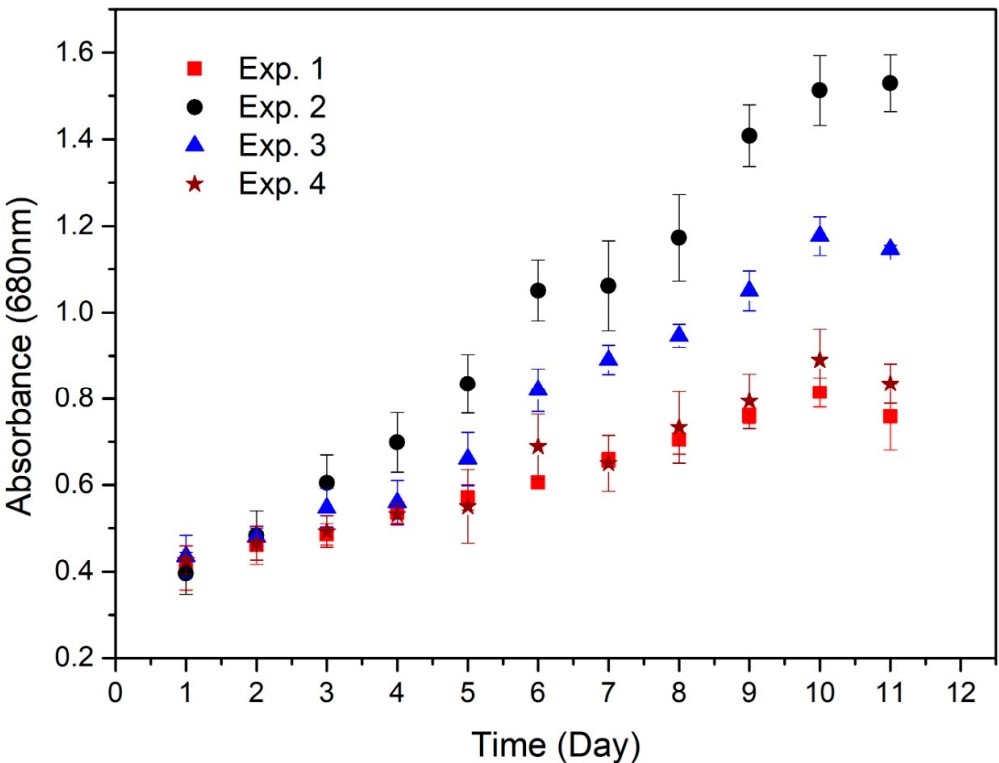

**Figure 1.** Growth profile of *Spirulina maxima* analyzed by the factors and levels studied.

Experiments 2 and 3 showed the highest cell development with maximal absorbances of $1.52 \pm 0.07$ and $1.15 \pm 0.01$, respectively. In both experiments, the light intensity was 54 $\mu$mol m$^{-2}$ s$^{-1}$, indicating that luminosity acts as a facilitator in the metabolization of the leachate by the cells. However, the percentage of the leachate in these experiments was different: 5% (*v/v*) in experiment 2 and 10% (*v/v*) in experiment 3. Experimental condition 1, performed with 5% (*v/v*) of leachate and 13.5 $\mu$mol m$^{-2}$ s$^{-1}$ of fluorescent light, presented the profile with the lowest growth rate, possibly due to the lower light incidence, even with a lower concentration of leachate in the medium. In experimental condition 4, performed with 10% (*v/v*) of leachate in 13.5 $\mu$mol m$^{-2}$ s$^{-1}$ of LED light, it also showed a low growth profile, indicating the significance of the light intensity for the assimilation of the leachate by the cell.

This obstacle to cell growth in a culture medium containing leachate is common in the literature. Liew et al. [22] observed that in cultures of *Chlorella vulgaris* with a high leachate content (89 to 100%) in the culture medium, cell growth was slower than those in cultures with a lower leachate concentration (33, 44, and 66%). Furthermore, the authors also observed that experiments with a lower percentage of leachate reached the maximum point of biomass productivity in a shorter time than experiments with a high leachate content. One explanation for this event is the presence of high concentrations of heavy metals when the percentage of leachate in the medium is high, causing toxicity and the inhibition of enzymatic activities, in addition to affecting chlorophyll synthesis, directly related to cell growth [23].

### 3.2. Factor Analysis of Productivities of Biomass, Carbohydrates, Phycocyanin, and Biochar

The results obtained, utilizing Taguchi's L4 arrangement, are presented in Table 4, while Table 5 displays the results of variance analysis (ANOVA) and the *F*-test. Analyzing Table 4, it becomes evident that high luminosity emerges as a common factor contributing to the highest yields of biomass, carbohydrates, and biochar. This result is corroborated by the ANOVA (Table 5, *F*-test). Figure 2 illustrates the effects of the response variables—biomass (a), carbohydrate (b), phycocyanin (c), and biochar (d) productivity—in relation to the

four parameters under examination for cultivation (leachate concentration, light source, and luminosity).

**Table 4.** Results of dry mass, carbohydrates, phycocyanin, and biochar productivities.

| Exp. | Factor | | | Productivity (mg $L^{-1}d^{-1}$) | | | |
|---|---|---|---|---|---|---|---|
| | LC | LS | Lum | Biomass | Carbohydrates | Phycocyanin | Biochar |
| 1 | 1 | 1 | 1 | 68.89 ± 2.83 | 6.38 ± 0.19 | 4.95 ± 0.60 | 24.16 ± 1.82 |
| 2 | 1 | 2 | 2 | 97.44 ± 3.20 | 12.82 ± 0.38 | 6.19 ± 1.54 | 34.79 ± 3.62 |
| 3 | 2 | 1 | 2 | 84.30 ± 2.08 | 7.73 ± 0.48 | 4.75 ± 0.08 | 28.32 ± 0.78 |
| 4 | 2 | 2 | 1 | 72.52 ± 2.33 | 5.63 ± 0.22 | 5.45 ± 0.97 | 24.02 ± 0.85 |

Note: LC—leachate concentration, LS—light source, and Lum—luminosity.

**Table 5.** Analysis of variance (ANOVA) for biomass, carbohydrates, phycocyanin, and biochar productivity obtained in the experiments.

| | Biomass | | | | | Carbohydrate | | | | |
|---|---|---|---|---|---|---|---|---|---|---|
| Factors | QSF | FD | AQSF | *F* | *p* | QSF | FD | AQSF | *F* | *p* |
| LC | 0.49174 | 1 | 0.49174 | 5.8988 | 0.041283 | 22.64121 | 1 | 22.64121 | 167.9292 | 0.000001 |
| LS | 2.18155 | 1 | 2.18155 | 26.1692 | 0.000912 | 8.25511 | 1 | 8.25511 | 61.2279 | 0.000051 |
| Lum | 14.01257 | 1 | 14.01257 | 168.0908 | 0.000001 | 58.06519 | 1 | 58.06519 | 430.6677 | 0.000000 |
| Residual | 0.66691 | 8 | 0.08336 | | | 1.07861 | 8 | 0.13483 | | |
| | Phycocyanin | | | | | Biochar | | | | |
| Factors | QSF | FD | AQSF | *F* | *p* | QSF | FD | AQSF | *F* | *p* |
| LC | 0.919344 | 1 | 0.919344 | 0.438901 | 0.543877 | 1.63075 | 1 | 1.63075 | 4.64895 | 0.097309 |
| LS | 4.367091 | 1 | 4.367091 | 2.084882 | 0.222257 | 1.49474 | 1 | 1.49474 | 4.26120 | 0.107929 |
| Lum | 0.262902 | 1 | 0.262902 | 0.125511 | 0.741019 | 10.52729 | 1 | 10.52729 | 30.01120 | 0.005405 |
| Residual | 8.378589 | 4 | 2.094647 | | | 1.40311 | 4 | 0.35078 | | |

Note: LC—leachate concentration, LS—light source, and Lum—luminosity; QSF = Quadratic Sum of Factors; FD = Degree of Freedom; AQSF = Average Quadratic Sum of Factors.

The highest productivity values obtained were as follows: biomass, 97.44 mg $L^{-1}d^{-1}$; carbohydrates, 12.82 mg $L^{-1}d^{-1}$; phycocyanin, 6.19 mg $L^{-1}d^{-1}$; and biochar, 34.79 mg $L^{-1}d^{-1}$. The light intensity in the photobioreactor was identified as the main factor for the increase in biomass concentration, with both highlighted in the effect graph (Figure 2) and the analysis of variance (ANOVA) (Table 5). All factors were statistically significant at 95% confidence, but the luminosity factor (C) proved to be the most influential for biomass yield, indicated by the high value of the *F*-test (173.86). The use of the high luminosity level (54 µmol $m^{-2}$ $s^{-1}$) maximized the dry biomass yield of the leachate cultivation according to the effect graph.

The decreasing order of relevance of the factors was as follows: luminosity (*p*-value = 0.0006, *F*-test = 30.085); light source, where the use of the high level (LED tube lamp) increased the yield in dry biomass; and leachate concentration (*p*-value = 0.0144, *F*-test = 9.6832), having a mild and negative impact on mass yield, possibly due to the higher turbidity of the culture medium at higher leachate concentrations.

The results in Table 5 show that the variation in the productivity of phycocyanin was mainly influenced by luminosity (factor C), which presented a greater amplitude of effect (Figure 2), whereas the type of light source had a less expressive influence. Although they were not statistically significant for the variation in phycocyanin (*p*-values for the factors: A = 0.82441; B = 0.44732; C = 0.13269) with 95% confidence, the *F*-test revealed that factor C had a greater prominent effect on phycocyanin production (*F*-test = 3.5489) compared to the other factors.

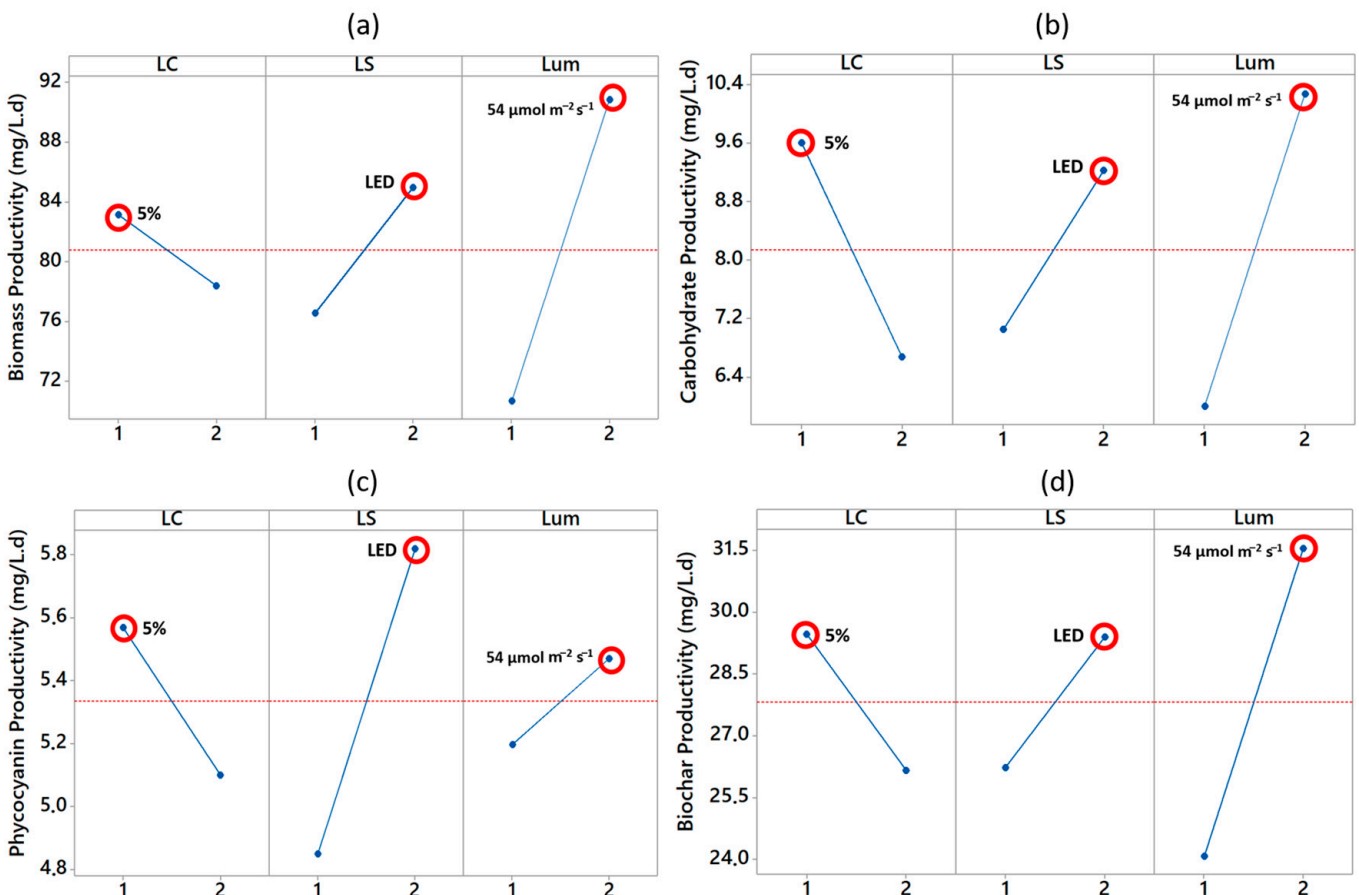

**Figure 2.** Main effect graphs on the response variables: (**a**) biomass productivity; (**b**) carbohydrate productivity; (**c**) phycocyanin productivity; (**d**) biochar productivity.

From Figure 2, the adjustment to maximize the productivity of phycocyanin suggests the leachate concentration used had little effect, suggesting the need for a high level of concentration (10% *v/v*) to consume the leachate in a greater quantity, a light source at a high level (LED tube lamp), and luminosity at a low level (13.5 μmol m$^{-2}$ s$^{-1}$), corresponding to experiment number 4 of Taguchi's L4 matrix. This combination did not require confirmatory experiments and was suggested based on observations of factors influencing phycocyanin production. This result is in line with the work carried out by Gomes et al. [24], who investigated the biomass production of the microalgae *Spirulina platensis* and the accumulation of phycocyanin, concluding that phycocyanin concentrations were higher when they utilized the modified *Venkataraman* medium.

The influence of luminosity and light sources on the cellular composition of *Spirulina* strains was also studied by Milia et al. [25]. The authors observed that microalgae cells demonstrated dependence on the type of lighting applied during cultivation. The orange light used in experiments enhanced biomass production (5973.3 mg L$^{-1}$), while white light influenced the increase in carbohydrate content (42.36 g 100 g$^{-1}$) and blue light acted to increase the phycocyanin fraction (14 g 100 g$^{-1}$). The authors also observed that white light caused a large production of biomass, thus reducing the availability of nitrogen in the cultivation medium. The reduction in nitrogen concentration causes the *Spirulina* cell to redirect energy towards the production of carbohydrates. Furthermore, the literature indicates that blue light positively influences the production of phycocyanin [26]. Research carried out by Park and Dinh [27] indicated that, when cultivating *Spirulina maxima* by applying red, white, and blue LEDs, the blue color stood out in increasing the content of phycobiliproteins (3.20 mg·g$^{-1}$ of phycocyanin; 0.19 mg·g$^{-1}$ of allophycocyanin; and

$0.97 \text{ mg} \cdot \text{g}^{-1}$ of phycoerythrin), while the red and white colors increased the concentration of chlorophyll $\alpha$.

The increase in carbohydrate productivity seems to have been influenced by the three factors, as shown in Figure 2 as well as in Table 5, which is confirmed by the ANOVA *p*-values being lower than 0.05 (confidence level above 95%) for all factors. However, the factors with the greatest impact on the production of reducing sugars were the leachate concentration (factor A) and luminosity (factor C), as evidenced by the higher value of the *F*-test (565.19 and 522.19, respectively) in comparison with factor B. The light source was also significant (*F*-test = 116.15 and $p = 0.000005$). Thus, the maximization of carbohydrates is favored using high levels of the light source (LED tube lamp) and luminosity ($54 \text{ }\mu\text{mol m}^{-2} \text{ s}^{-1}$), combined with the application of a low level of leachate concentration (5% *v/v*). It was noted that maximizing carbohydrates aligns with a specific combination of factors: a low leachate concentration and high levels of both the light source and luminosity. This set was particularly evident in experiment number 2 of Taguchi's L4 matrix, eliminating the necessity for a confirmatory experiment.

Rempel et al. [14] also studied the influence of leachate as a component of the culture medium on the cellular composition of *Spirulina platensis* and *Scenedesmus obliquus*. Both species were cultivated using fractions of 5 and 10% of leachate incorporated into the culture medium, a controlled temperature of 25 °C, and LED lighting of $138.3 \text{ }\mu\text{mol m}^{-2} \text{ s}^{-1}$. The authors identified that *Spirulina* presented the highest percentage of carbohydrates (16.47%) in the culture medium containing 10% leachate, while *Scenedesmus* reached a value of 18.06% in the culture medium containing the same percentage of leachate. The authors pointed out a relationship between the increase in carbohydrate content and the reduction in the concentration of nutrients available in the environment. With an abundance of nutrients, cyanobacteria have a high protein content; however, this content can be modified when there is a nutrient restriction as the cell begins to convert proteins into carbohydrates, as this biocompound represents the cell's main energy reserve [28].

From Figure 2, it can be observed that the cultivation with the leachate concentration (factor A) had the greatest amplitude of effect on the biochar yield, whereas the other factors (B and C) had little influence on this result. Despite this, the ANOVA analysis did not show a significant confidence level (*p*-values for the factors: A = 0.39307; B = 0.89359; C = 0.71064) due to inconsistency in the repeatability of the cultures performed. The *F*-test, however, highlights that factor A had the greatest effect on the biochar yield (*F*-test = 0.914526), this being the most relevant parameter compared to the other factors. The application of the low-level leachate concentration (5% *v/v*) maximized the percentage of biochar obtained after thermochemical treatment. Factors B and C did not show a significant influence on the biochar yield as their effects were not very expressive in the effect graph.

These factors did not interfere with the composition of the biomass; only the leachate concentration influenced the generation of volatile or nonvolatile biocompounds during the thermal conversion process. The lower concentration of leachate favored the formation of nonvolatile biocompounds in the biomass, optimizing the biochar yield during pyrolysis. To maximize the percentage of biochar, the suggested combination of factors is as follows: leachate concentration at a low level (5% *v/v*), light source at a high level (LED tube lamp), and luminosity at a high level ($54 \text{ }\mu\text{mol m}^{-2} \text{ s}^{-1}$). This adjustment was obtained from experiment number 2 of Taguchi's L4 matrix, thus not requiring confirmatory experiments due to the observations of the factors that impact the biochar yield.

In a study by Yu et al. [29], the researchers achieved a peak biochar content of 38.4% and a productivity of $334.08 \text{ mg L}^{-1} \text{ d}^{-1}$ by cultivating *Chlorella vulgaris* in a batch reactor and introducing $CO_2$ through three different flows. They attributed these values to their pyrolysis methodology, exposing samples to 500 °C for 30 min. In contrast, the pyrolysis process applied in this work was gradual, reaching a maximum temperature of 310 °C, yielding results allied with those reported by the authors.

In the work conducted by Zhu and Zou [30], the authors evaluated the adsorption potential of methyl orange dye on biochar produced from *Spirulina* residue and the effect

of pyrolysis time on its absorption capacity. The authors observed that with increasing pyrolysis time, the surface roughness of the biochar increased, changing its physical-chemical properties and decreasing the adsorption capacity.

Binda et al. [4] also estimated biochar production efficiency through prolonged pyrolysis at lower temperatures. They conducted pyrolysis at 350 °C for 1 h, obtaining biochar levels for three microalgae species: *Chlorella vulgaris* (35%), *Nannochloropsis* sp. (42%), and *Spirulina* sp. (37%). Comparing these results with the maximum biochar content obtained in this work, it is evident that the method can produce substantial biochar levels at lower temperatures, therefore contributing to energy conservation efforts.

The analysis is in line with previous studies that indicate the potential of landfill leachate as a growing medium, due to the nutrients present, but it also warns of the need for dilution due to potentially toxic components. The suggested adjustment to maximize the dry mass yield (g/L) involves the use of a low leachate concentration, LED tube lamp light source, and high luminous intensity ($54\ \mu\text{mol}\ \text{m}^{-2}\ \text{s}^{-1}$), which is the experimental condition number 2 of Taguchi's L4 matrix, eliminating the need for a confirmatory experiment.

## 4. Conclusions

The results showed that *Spirulina maxima* presented good development in Zarrouk's medium supplemented at the lowest level of leachate concentration (5% *v/v*), resulting in a maximum yield of $97.44 \pm 3$ ($\text{mg}\ \text{L}^{-1}\ \text{d}^{-1}$) for dry biomass, $12.82 \pm 0.38$ ($\text{mg}\ \text{L}^{-1}\ \text{d}^{-1}$) for carbohydrates, and $34.79 \pm 3.62$ ($\text{mg}\ \text{L}^{-1}\ \text{d}^{-1}$) for biochar. The adoption of the LED tube lamp is beneficial for most productions of biocompounds such as dry mass, carbohydrates, and phycocyanin due to the emission of a wavelength range that is more favorable to cell development. It was observed that the production of dry matter and carbohydrates was maximized with the luminosity of $54\ \mu\text{mol}\ \text{m}^{-2}\ \text{s}^{-1}$, whereas the lowest luminosity of $13.5\ \mu\text{mol}\ \text{m}^{-2}\ \text{s}^{-1}$ favored the maximum production of phycocyanin ($6.19 \pm 1.54\ \text{mg}\ \text{L}^{-1}\ \text{d}^{-1}$). Thus, the use of leachate as a substitute for macronutrients in Zarrouk's medium for the cultivation of *Spirulina maxima* has become a viable alternative in the production of biocompounds as long as it is used at the appropriate level, mitigating, therefore, the impact of discarded leachate on nature.

**Supplementary Materials:** The following supporting information can be downloaded at: https://www.mdpi.com/article/10.3390/agriengineering6020074/s1, Figure S1: DNS solution calibration curve for acid-labile carbohydrate concentration in *Spirulina maxima*; Table S1: Zarrouk (1966) medium composition.

**Author Contributions:** D.H.P.G. conceptualized and designed the experiments, supervised the work, and reviewed and edited the manuscript. G.V.T. supervised the work and set up the airlift bioreactors for the experiments. M.L.d.S. conducted the experiments and prepared the original draft of the manuscript. W.R.d.S. supervised the conduction of the experiments, revised the discussion of the results, and reviewed and edited the manuscript. A.L.G.F. revised the discussion of the results and reviewed and edited the manuscript. All authors have read and agreed to the published version of the manuscript.

**Funding:** This research was funded by the São Paulo State Research Foundation (FAPESP, grant number 2019/21901-2).

**Data Availability Statement:** The data presented in this study are available on request from the corresponding author.

**Acknowledgments:** The authors are thankful to the São Paulo State Research Foundation (FAPESP, grant no. 2019/21901-2) and the Brazilian Federal Agency for Support and Evaluation of Graduate Education (CAPES, finance code 001) for their financial support.

**Conflicts of Interest:** The authors declare no conflicts of interest.

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
