# Peer review of "The Cultivation of Spirulina maxima in a Medium Supplemented with Leachate for the Production of Biocompounds: Phycocyanin, Carbohydrates, and Biochar"

_agriengineering, doi:10.3390/agriengineering6020074_

Round 1
Reviewer 1 Report
Comments and Suggestions for Authors
The manuscript focuses on the cultivation of Spirulina maxima (Setchell & N.L.Gardner) Geitler 1932 in medium supplemented with leachate. Landfill leachate studies are very relevant and popular nowadays. Therefore, experimental work on cultivation of microorganisms using leachate is very valuable. I appreciate the design of the experiment and its description in the Materials and Methods section.
But I have some questions about the Results and Discussion sections. First, the results are described very poorly, while the Discussion section mainly describes the results obtained during the experiments. Therefore, in my opinion, it is necessary either to combine these sections into one "Results and Discussion" or to move the description of the results from the Discussion. For example, L. 35-36 is a classical description of the experimental results. The Results section mainly presents graphs without sufficiently describing them. For example, in Figure 1, there is no explanation of what ABS is. In the caption to the graph it would be good to explain, as this abbreviation occurs only once in the manuscript. On the other hand, very little literature on the topic is cited in the Discussion section.
In addition, I would like to note that at the first mention the species name it is recommended to indicate the author and year. This is especially important if the authors leave the old species name rather than giving the new one, which is given at https://www.algaebase.org/. The same applies to the first mention of the genus Spirulina.
Considering the value of such experimental work, I believe that after revision of the Results and Discussion sections by the authors, this manuscript will be suitable for publication in AgriEngineering.
Author Response
Dear Reviewer #1, responses to your comments can be found in the attached file.
Best regards.

Reviewer 2 Report
Comments and Suggestions for Authors
1. Introduction part: give some research examples that microorganisms can be grow rapidly in the harshest aquatic environments.
2. Materials and Methods part: Provide the detailed information of the leachate from the landfill of the municipality.
3. Results part. Only figures and tables are provided, there were no description can be found.
4. Discussion part: Discuss whether the components in this leachate are common, whether this approach can be used in other industry leachate.
Author Response
Dear Reviewer #2, responses to your comments can be found in the attached file.
Best regards.

Reviewer 3 Report
Comments and Suggestions for Authors
Thank you for the opportunity to review this paper.
The identification of new methods to obtain large quantities of spirulina and bioactive compounds is a challenge for many groups of researchers, considering the multiple uses of spirulina. Also, identifying methods that do not have negative effects on the environment are all the more important.
In the Materials and Methods chapter, the study methods proposed by the authors are presented, methods that were chosen correctly and in accordance with the purpose of the work. The parameters determined for productivity assessment were carefully selected.
The obtained results are presented in detail; the statistical analysis of the obtained data is well documented. The presentation of the research results was made through tables and figures that clearly explain the efficiency of each experimental method used.
The Discussion chapter reports its own results to existing data in the literature.
The manuscript is presented clearly and coherently, it is well scientifically documented. Recent bibliographic sources have been carefully supplemented by older sources of information.
The conclusions established by the authors are clear, the conducted study provides a scientific basis for new studies and the applicability of the results obtained depending on the experimental method used.
I recommend that this paper be accepted and published in this journal.

Author Response
Dear Reviewer #3, responses to your comments can be found in the attached file.
Best regards.

Reviewer 4 Report
Comments and Suggestions for Authors
This manuscript appears to be in an initial phase of development. There is significant need for a comprehensive revision and enhancement in terms of both the scope and methodology. The adoption of landfill leachate as an additive in Spirulina's growth medium presents a novel approach but lacks an in-depth investigation into the potential risks or setbacks, such as inconsistencies in the composition of the leachate and subsequent implications on downstream processing.
The title implies that phycocyanin, carbohydrates, and biochar are the primary by-products of this procedure. It would be of significant improvement if the authors expound on their decision to focus on these specific Spirulina compounds and whether exploration into or exclusion of other compounds was considered. The term "Biofuels" shows up in the keywords; how does it relate to Spirulina?
The manuscript would also benefit greatly from a detailed reasoning behind the selection of the light source and intensity. The varying results found with different light intensities prompt an important question - how would the most fitting intensity be chosen for practical applications? Moreover, “lux”, the unit of light intensity employed here, is generally discouraged in cyanobacteria-related research.
The reviewer is of the opinion that this manuscript, in its current form, doesn't meet the acceptable standards for publication in this journal.
Comments on the Quality of English LanguageExtensive editing of English language required.
Author Response
Dear Reviewer #4, responses to your comments can be found in the attached file.
Best regards.

Round 2
Reviewer 1 Report
Comments and Suggestions for Authors
After reviewing the revised version of the manuscript "Cultivation of Spirulina maxima in medium supplemented with leachate for the production of biocompounds: phycocyanin, carbohydrates and biochar", I really appreciated the additions to the Introduction section. The Results and Discussion section looks much better after the redesign, the text is clearer and more complete. In addition, the authors have added a lot of literature sources.
The only thing left to correct:
L. 80 is the first mention of Spirulina. At my request, the authors of the species have been inserted, but they should not be italicized. In addition, after the year, “:923”, accidentally copied from the algaebase, should be removed.
After this small correction, the article will be ready for publication.
Author Response
The new response to reviewer #1 is attached.

Reviewer 4 Report
Comments and Suggestions for Authors
The author has addressed the majority of the concerns raised by this reviewer. As for the unit of light intensity, it is noteworthy that the articles listed by the authors appear to be focused on microalgae rather than cyanobacteria.
Comments on the Quality of English LanguageModerate editing of English language required.
Author Response
The new response to reviewer #4 is attached.
